# ATP Analogues for Structural Investigations: Case Studies of a DnaB Helicase and an ABC Transporter

**DOI:** 10.3390/molecules25225268

**Published:** 2020-11-12

**Authors:** Denis Lacabanne, Thomas Wiegand, Nino Wili, Maria I. Kozlova, Riccardo Cadalbert, Daniel Klose, Armen Y. Mulkidjanian, Beat H. Meier, Anja Böckmann

**Affiliations:** 1Laboratory of Physical Chemistry, ETH Zurich, 8093 Zurich, Switzerland; denis.lacabanne@mrc-mbu.cam.ac.uk (D.L.); nino.wili@phys.chem.ethz.ch (N.W.); Riccardo.Cadalbert@nmr.phys.chem.ethz.ch (R.C.); daniel.klose@phys.chem.ethz.ch (D.K.); 2Medical Research Council Mitochondrial Biology Unit University of Cambridge, Cambridge Biomedical Campus, Keith Peters Building, Hills Road, Cambridge CB2 0XY, UK; 3Department of Physics, Osnabrueck University, 49069 Osnabrueck, Germany; makozlova@uni-osnabrueck.de (M.I.K.); armen.mulkidjanian@uni-osnabrueck.de (A.Y.M.); 4School of Bioengineering and Bioinformatics and Belozersky Institute of Physico-Chemical Biology, Lomonosov Moscow State University, 119234 Moscow, Russia; 5Molecular Microbiology and Structural Biochemistry UMR 5086 CNRS/Université de Lyon, Labex Ecofect, 69367 Lyon, France

**Keywords:** solid-state NMR, ELDOR-detected NMR, ATP hydrolysis, ATP analogues, DnaB helicase, ABC transporter

## Abstract

Nucleoside triphosphates (NTPs) are used as chemical energy source in a variety of cell systems. Structural snapshots along the NTP hydrolysis reaction coordinate are typically obtained by adding stable, nonhydrolyzable adenosine triphosphate (ATP) -analogues to the proteins, with the goal to arrest a state that mimics as closely as possible a physiologically relevant state, e.g., the pre-hydrolytic, transition and post-hydrolytic states. We here present the lessons learned on two distinct ATPases on the best use and unexpected pitfalls observed for different analogues. The proteins investigated are the bacterial DnaB helicase from *Helicobacter pylori* and the multidrug ATP binding cassette (ABC) transporter BmrA from *Bacillus subtilis*, both belonging to the same division of P-loop fold NTPases. We review the magnetic-resonance strategies which can be of use to probe the binding of the ATP-mimics, and present carbon-13, phosphorus-31, and vanadium-51 solid-state nuclear magnetic resonance (NMR) spectra of the proteins or the bound molecules to unravel conformational and dynamic changes upon binding of the ATP-mimics. Electron paramagnetic resonance (EPR), and in particular W-band electron-electron double resonance (ELDOR)-detected NMR, is of complementary use to assess binding of vanadate. We discuss which analogues best mimic the different hydrolysis states for the DnaB helicase and the ABC transporter BmrA. These might be relevant also to structural and functional studies of other NTPases.

## 1. Introduction

Nucleosides triphosphates (NTPs), such as ATP (adenosine triphosphate) and GTP (guanosine triphosphate), are used as energy source or as allosteric effector by a number of proteins, involved for instance, in metabolism, active transport, cell division or DNA/RNA synthesis. However, the mechanism of NTP hydrolysis in proteins are still poorly understood, especially its coupling to functional events, such as movement of proteins along nucleic acids. Indeed, detailed mechanistic insight is lacking for a number of systems, including even intensively studied systems such as dyneins [1], ABC importers [2]/exporters [3] or DNA helicases [4]. Experimentally, catching the events that occur during the NTP hydrolysis is highly challenging. Structural techniques such as X-ray crystallography, cryo-electron microscopy (cryo-EM) and nuclear magnetic resonance (NMR) mainly provide static snapshots of protein states of typically highly complex reaction coordinates of biomolecular reactions. These can then be combined with molecular dynamics simulations (MD) to obtain further information about the dynamics of such processes and to establish the chronological sequence [3,5,6,7,8]. In this context, it is highly desirable to better investigate how ATP/GTP analogues, usually with a modified or replaced γ-phosphate group, can mimic the intermediate catalytic states in order to obtain relevant snapshots of the reactions involving NTP hydrolysis. Indeed, it is well known that mimics can never fully represent naturally occurring states, as the modifications of NTPs change their conformation as well as their chemical properties other than their tendency to hydrolyse. While it is important to use NTP mimics described to be strongly hydrolysis-resistant, the true hydrolysis state must often be confirmed experimentally.

The choice of the adequate analogue is thus of importance in structural studies, but guidelines are sparse and can be highly protein-dependent. We herein focus on analogues often used to access three important states of ATP-hydrolysis: the pre-hydrolytic state, the transition state and the post-hydrolytic state (see Scheme 1A for the artificial ATP hydrolysis scheme highlighting analogues used to mimic the different states). We describe how NMR and EPR can be used to gain detailed information on the analogue used and the conformational and dynamic state it induces in the protein. We investigate this for two proteins, the bacterial DnaB helicase from *Helicobacter pylori* involved in DNA replication, and an ABC transporter implicated in multidrug resistance, BmrA (*Bacillus subtilis* multidrug resistance ATP binding cassette transporter), which share high similarities in their ATP binding sites [9]. Solid-state NMR and EPR are highly suitable to study large, noncrystalline protein assemblies, which are represented by DnaB and BmrA. The proteins are, in their multimeric states and, for BmrA, embedded in a *Bacillus subtilis* lipid membrane, sedimented directly into the solid-state NMR rotor in an external ultracentrifuge [10], a sample preparation approach that allows for the study of the investigated analogues. The protein samples prepared by this approach are highly concentrated in the NMR rotor (protein concentration of around 400 mg/mL), and have been shown to be stable over several years [11]. A description of the NMR techniques developed to investigate such molecular machines is given in detail in reference [12].

Figure 1 and Table 1 summarize the most important NMR experiments and the nomenclature used herein and gives the information content of NMR spectra and the underlying NMR observables. The standard experiment to establish a chemical-shift fingerprint of the protein is the ^13^C-^13^C DARR, a two-dimensional correlation experiment using the dipolar assisted rotational recoupling (DARR) scheme [13,14]. Besides delivering a first sample quality check (Figure 1A), isolated peaks in such spectra, often found in the alanine or threonine regions, can serve to follow the conformational changes along the reaction coordinate. Differences in the cross-peak positions (encoding the chemical shift) in such spectra characterize the different protein conformations, produced by incubating the protein with ligands. Such changes are denoted as chemical-shift perturbations (CSPs) (Figure 1B). Additionally, appearing or disappearing resonances might be observed in the spectra, pointing to dynamic changes of the protein (Figure 1C). ^31^P NMR experiments allow for direct detection of nucleotides, such as ATP mimics or DNA/RNA [12] (Figure 1D). The ^31^P chemical-shift values react very sensitively to small conformational changes, e.g., in the phosphate backbone of ATP mimics. ^31^P direct-pulsed experiments (recorded with short repetition times, Figure 1E) are used to detect unbound nucleotides present in the water phase in contact with the protein (interacting water) or the supernatant of the NMR rotor [15]. ^31^P cross-polarization (CP) based experiments are employed to detect immobilized nucleotides, particularly those bound to the protein (Figure 1E). ELDOR-detected NMR (EDNMR) is a pulsed EPR-technique and allows for the measurement of hyperfine couplings of paramagnetic spin centers to nearby spin-active nuclei [16,17,18,19]. We herein use this technique to detect NMR-active nuclei in the vicinity of the ATP-cofactor (for this the diamagnetic Mg^2+^ has to be substituted by paramagnetic Mn^2+^), particularly focusing on ^31^P and ^51^V nuclear spins [20]. If a ^51^V nucleus is in spatial proximity to the Mn^2+^ ions, the ^51^V resonance should be detected in the EDNMR spectra (Figure 1F). Note that the hyperfine coupling to the ^51^V is often not resolved, in contrast to the ^31^P couplings.

We here make use of these magnetic-resonance approaches to study the states of DnaB and BmrA induced by phosphate-modified NTP analogues widely used to mimic the three major states of the NTP hydrolysis reaction and report on the efficiency of the analogues to actually mimic the desired states.

## 2. The Different Hydrolysis States and the ATP-Mimics Used to Induce Them

The pre-hydrolytic state, where ATP is bound to the protein, is often already associated with protein conformational changes [21,22,23,24]. In this state, the γ-phosphate adopts a tetrahedral geometry (note that a similar discussion also holds for GTP). This geometry can change to a trigonal-bipyramidal geometry generating a pentavalent terminal phosphate group [25,26]. Most of the analogues are mimicking the pre-hydrolytic state because their γ-phosphate (in case of nonhydrolyzable analogues) or the γ-phosphate-mimicking group adopts a tetrahedral geometry [27]. The most commonly used nonhydrolyzable analogues are: AMPPNP (adenylyl imidodiphosphate) [28], AMPPCP (adenylyl methylenediphosphate) [29], AMPCPP (alpha,beta-methylene-triphosphonate) and ATP-γ-S (adenosine 5’-(gamma-thiotriphosphate)) [30]. In addition, the pre-hydrolytic state appears to be mimicked by ADP-BeF_x_ [27] (Figure 2 and Figure 3A). BeF_x_ forms a strictly tetrahedral complex (specific to the pre-hydrolytic state); a penta-coordinated bipyramidal geometry (describing the transition state, see below) is excluded in this case [26,27]. The nonhydrolyzable ATP analogues are not completely resistant to hydrolysis. While the rate of hydrolysis of these analogues is indeed significantly lower, several of them can still be hydrolysed by many ATPases [31,32,33,34,35,36,37,38]. This behaviour can differ from protein to protein, and an analogue can fail to mimic a pre-hydrolytic state or may mimic a different/uncomplete pre-hydrolytic state in certain cases [26,39,40,41]. This difference can be observed between distinct protein families, or even within the same family [42], as shown in this work for the two model systems discussed.

The transition state (the “ATP-is-ready-to-be-split” state) can be accessed by an associative and dissociative mechanism, which represent the two extreme cases discussed in the literature [43,44,45,46,47,48]. In the case of an associative mechanism, the phosphorus possesses a pentavalent geometry. The nucleophilic attack of a water molecule at the γ-phosphate, forming a H_2_O-P bond, in this scenario occurs before the leaving group departs and before the P-O bond breaks (similar to a S_N2_ nucleophilic substitution, see Scheme 1B,C). In contrast, in the case of a dissociative mechanism, the nucleophilic attack of a water molecule at the γ-phosphate occurs after the leaving group was released, generating a metaphosphate intermediate before it collapses onto the acceptor nucleophile (similar to a S_N1_ reaction). The transition state can be simulated by employing three prominent mimic groups in combination with ATP or ADP: aluminium fluoride (ADP:AlF_x_) [27,49,50], magnesium fluoride (ADP:MgF_x_) [51], and vanadate (ADP:Vi) [52]. In some enzymes, ATP hydrolysis is required prior to the binding of the transition-state mimic [53,54]. In structural studies, aluminium fluoride is most frequently used as a mimic of the γ-phosphate in the transition state, as evidenced by analysing the number of deposited structures in the PDB database (Figure 2A). When the analogue in the presence of ADP is complexed with the protein, ADP:AlF_x_ is believed to mimic the transition state of an ATP molecule. Two configurations of this analogue have been observed: ADP:AlF_3_ and ADP:AlF_4_^−^. In the ADP:AlF_4_^−^ mimic (two-thirds of in the PDB deposited AlF_x_-containing structures) the AlF_4_^−^ group is in a squared-planar geometry and forms an octahedral complex with two oxygen ligands in the apical positions. While one ligand is provided by the β-phosphate, the other ligand comes from the hydrolytic water molecule in the attack position next to the phosphorus atom. It is believed that such a structure mimics the interaction of the catalytic water molecule with the γ-phosphate in the anionic transition state for phosphoryl transfer [49,55]. AlF_3_ (one-third of in the PDB deposited AlF_x_-containing structures) is in a trigonal-planar geometry forming a bipyramidal complex resembling the geometry of the transition state [49,55].

MgF_3_^−^ shows nearly the same geometry as AlF_3_ but carries a negative charge similarly to the anionic γ-phosphate in the transition state. AlF_3_ and MgF_3_^−^ are structurally similar and have similar scattering factors for X-rays; therefore, it has been suggested that MgF_3_^−^ is present in some crystal structures, which are indicated as containing NDP:AlF_3_ [56]. Indeed, Mg^2+^ ions are usually present in the samples as cofactors of NTP hydrolysis. In contrast to X-ray, NMR can differentiate the two metal fluorides so that in few cases, the presence of MgF_3_^−^ in the active site was directly shown [56,57,58]. For more information about such cases and the use of metal fluorides as ATP or phosphate analogues, we refer the reader to the two comprehensive recent reviews [55,58]. Finally, vanadate-containing ATP:Vi or ADP:Vi are used as a transition-state mimic for a variety of proteins. Vanadate is an oxoanion of vanadium which shares structural and chemical similarities with phosphate molecules mimicking the hydrolysis transition state [52,59]. It is known that the simple form of the oxoanion (VO_4_^3−^) can adopt a penta-coordinated, trigonal bipyramidal geometry around the central vanadium in presence of ADP [60]. These properties make the vanadate a phosphate mimic of the transition state for phosphoryl transfer so that vanadate acts as an inhibitor for some ATPases. As previously described by Davies et al. [52], vanadate can be used to mimic phosphoryl transfer, and structures of different protein families including myosin [61,62], dynein [63], kinesin [40], ABC transporters [60,64,65], heat shock protein (Hsp70s) [66], NS3 helicase (dengue virus) [67], nucleoside-diphosphate kinase [68] or F1-ATPase [69] are reported. The main advantage of vanadate is that it can form covalent bonds with the oxygens of phosphate groups from ADP or other ligands [52]. Interestingly, this is not always the case [68], as there are structures where a vanadate is not bound to ADP, but still stabilizes the transition state. It is also noteworthy that vanadate does not work as an inhibitor or as transition-state mimic for all proteins with ATPase activity [70].

Finally, the post-hydrolytic state corresponds to a situation where the nucleotide diphosphate and the previously associated γ-phosphate are separated, but both are still bound to the protein, or, alternatively, where the γ-phosphate is already released, and only ADP is bound to the protein. The post-hydrolytic state where the γ-phosphate is not released can be mimicked not only by an orthophosphate [67,71,72] but also by a sulphate ion, SO_4_^2−^ [71,73,74,75]. Note that sulphate ions have only two ionisable oxygens (with pK_a_ below 2) [76].

The overall conformational variability of NTP analogues can be seen by overlaying the structures extracted from the PDB and by aligning them on their nucleoside parts (Figure 3). AMPPNP and ATP-γ-S adopt a wider range of conformations (Figure 3A,C) than AMPPCP and ADP:BeF_x_ (Figure 3B,E), although this allows for a qualitative statement only, since the total numbers of deposited structures in the PDB are different (see Figure 2). AMPPNP and ATP-γ-S thus seem to adapt their conformation to the protein-binding pocket, while for AMPPCP and ADP:BeF_x_ it may be the protein that adapts. For the transition-state analogues, it is difficult to make the same comparison due to the small number of structures available. However, ADP:AlF_x_ shows a significant distribution of structures as well (Figure 3F).

In sum, from the eight mainly used analogues for structural studies, five are used to mimic the pre-hydrolytic state: AMPPNP, AMPPCP, ATP-γ-S, AMPCPP and ADP:BeF_x_, three to mimic the transition state: ADP:AlF_x_, ADP:MgF_x_ and ADP:V_i_, and ADP and ADP:SO_4_^2−^ to mimic the post-hydrolytic state. Note that also other NTP analogues exist that differ structurally through the introduction of atoms or groups (e.g., fluorescent probes, biotin groups, etc.) on the base, sugar, or triphosphate regions of the molecule [77,78,79]. A complete overview is given in reference [77].

## 3. The DnaB Helicase and the ABC Transporter BmrA

The usefulness of particular ATP mimics for structural studies strongly depends on the nature of the protein of interest, as shown in Figure 2B for the example of DNA helicases and ABC transporters. The two proteins were subject to studies in the last years in our laboratories: the bacterial helicase DnaB from *Helicobacter pylori* [38,80,81,82,83,84,85,86,87,88] and the ABC transporter BmrA from *Bacillus subtilis* [86,89,90,91,92]. In the presence of double-stranded DNA, the DnaB from *Helicobacter pylori* is a double-homo hexamer of 59 kDa monomers with each hexamer moving along its single DNA strand, whereas BmrA from *Bacillus subtilis* is a dimeric membrane protein of 65 kDa monomers. The two proteins are well-characterized ATP-fuelled proteins. In both proteins, the chemical energy released during ATP hydrolysis in the nucleotide-binding domain (NBD) is converted into mechanical work, which, e.g., enables the movement of DnaB along a double-stranded DNA and its unzipping, as well as the transportation of molecules across the membrane by ABC transporters. Both proteins belong to the vast family of P-loop fold NTPases, one of the largest protein superfamilies. In any genome 10–20% of proteins code for P-loop fold domains [93,94,95]. P-loop fold NTPases are characterized by their signature GxxxxGK [S/T] sequence motif, also known as the Walker A motif [96]. This motif is responsible for binding the triphosphate chain and is often called the P-loop (phosphate-binding loop) motif [97]. In the P-loop fold, the conserved Lys residue forms hydrogen bonds with the β- and γ-phosphate groups of ATP or GTP. Another conserved motif, known as the Walker B motif, is composed of four hydrophobic residues ended by an aspartate residue. The conserved Asp residue stabilizes the metal ion cofactor Mg^2+^ [96].

The *C*-terminal NBD of DnaB belongs to the superfamily 4 (SF4) of helicases, which in turn belongs to the class “RecA and F_1_/F_O_-related ATPases” (hereafter abbreviated as RecA/F_1_-related ATPases) of P-loop old NTPases. The ABC transporter BmrA belongs to a separate class of ABC transporters [93,94,95]. Both the RecA/F_1_-related ATPases and ABC transporters belong to the ASCE division of P-loop fold NTPases. The members of this division are characterized by an additional β-strand in the P-loop fold and a catalytic glutamate (E) residue next to the attacking water molecule [94,95,98]. The glutamate residue stabilizes the catalytic water molecule and, perhaps, operates as a catalytic base for ATP hydrolysis [99].

To avoid a futile NTP hydrolysis, P-loop fold NTPases are initiated before each turnover by activating moieties provided either by other proteins or by domains of the same protein [100,101,102,103,104]. The activating moiety interacts with the triphosphate chain and triggers the hydrolysis. The ATP hydrolysis in DnaB is induced by an interaction with an arginine residue that is provided by the neighbouring subunit of the same oligomer [105,106]. In ABC transporters, one of the NBDs is believed to activate hydrolysis within the active site in the other NBD by providing a signature LSGGQ motif [64,106].

Two analogues were mainly used in structural studies of helicases and ABC transporters (Figure 2): AMPPNP and ATP-γ-S, which both mimic the pre-hydrolytic state. The transition state is mainly mimicked by ADP:AlF_x_ for the helicases, and ADP:Vi for the ABC transporters. Regarding the literature, this state is underrepresented compared to the pre-hydrolytic state.

We here gather information from published experiments, as well as present complementary original data, in order to give a compilation of ATP analogues and their mimicking power for the two proteins DnaB and BmrA, as assessed by magnetic-resonance methods, namely NMR and EPR.

## 4. Results and Discussion

### 4.1. The Pre-Hydrolytic State Mimicked by AMPPCP, AMPPNP and ATP-γ-S

In order to characterize the pre-hydrolytic state, we first investigated DnaB and BmrA in the presence of AMPPNP, AMPPCP, and ATP-γ-S. It however appeared that ATP-γ-S was completely hydrolysed during the rotor filling by BmrA (one hour of filling) and DnaB (overnight filling), as monitored by ^31^P solid-state NMR experiments (see Appendix A), and was thus of no further use. We therefore focused on AMPPNP and AMPPCP. Since a major function of the DnaB helicase is to bind to DNA, protein samples were also prepared with the ATP analogue and single-stranded DNA (here a DNA-fragment of 20 thymidine nucleotides abbreviated as (dT)_20_). The presence of three signals in the 1D CP ^31^P NMR spectrum (Figure 4A, left panel) indicates binding of the triphosphate AMPPCP to DnaB. However, the resonances of the phosphorus α and β are rather broad. This broadening might indicate inhomogeneities in the binding site in the environment of the ligand, or chemical-exchange broadening effects. In contrast, in the presence of DNA and AMPPCP, the ^31^P resonances in the 1D CP ^31^P spectrum are very sharp (Figure 4B, left panel). This indicates that the DnaB:DNA complex fixes AMPPCP with high homogeneity.

The 2D ^13^C-^13^C DARR experiments recorded on DnaB:AMPPCP show not only chemical-shift perturbations when compared to the apo protein, but also dynamic changes, as can be seen in the extract of the alanine region (Figure 4A, right panel) by the disappearance of resonances, which could be assigned to the *N*-terminal domain [82], which is important for binding the DnaG primase within the primosome. As illustrated by the equivalent 2D ^13^C-^13^C DARR experiment on the DNA-bound DnaB (Figure 4B, right panel), the binding of AMPPCP induces stronger CSPs due to larger conformational changes of the protein, but no dynamic effects of the *N*-terminal domain were observed.

In principle, AMPPNP and AMPPCP should have a similar effect on DnaB, as both should induce the pre-hydrolytic state. However, it is clear from the NMR spectra that the effects of these two analogues are very different. First, as highlighted by the 1D CP ^31^P spectrum (Figure 4C, left panel), the presence of multiple resonances from the phosphate groups of AMPPNP indicates several structurally slightly different bound AMPPNP molecules. Interestingly, the 2D ^13^C-^13^C DARR spectrum reveals that the disappearance of the *N*-terminal domain resonances upon binding of AMPPCP is not observed in case of AMPPNP (Figure 4C, right panel). Also, we had observed that in presence of DNA, all AMPPNP is hydrolysed by the helicase [38]. Consequently, as shown by Figure 4D right panel, no AMPPNP is bound to the protein when DNA binds to the helicase, and the 2D ^13^C-^13^C DARR spectrum of DnaB in the presence of AMPPNP looks highly similar to DnaB without the analogue, which is not detected in the ^31^P spectra either (Figure 4D left panel).

BmrA also binds AMPPCP, as shown by the 1D CP ^31^P spectrum (Figure 4E, left panel). However, the rate of AMPPCP hydrolysis is much higher, compared to DnaB, and degradation products of AMPPCP can be observed already four hours after the rotor filling in the supernatant of the NMR rotor, as shown in Figure 4E right panel (see red stars in the Figure). We recorded a 2D ^13^C-^13^C DARR experiment of BmrA:AMPPCP (two days of acquisition), and the spectrum is virtually the same as the one of BmrA in the apo state. Possibly, AMPPCP has been rapidly hydrolysed, and an insufficient amount of AMPPCP only remained bound on BmrA. Similar to AMPPCP, AMPPNP binds to BmrA (Figure 4F), but was also rapidly hydrolysed (data not shown). The analysis of ^31^P NMR spectra for protein samples containing lipids or DNA is more difficult due to the overlap between the ^31^P γ- and β-phosphate signals from AMPPNP and those from lipid/DNA.

To overcome the hydrolysis problem with BmrA and to obtain a snapshot of the protein in its pre-hydrolytic state, we used an alternative approach, which is based on using mutant forms of the protein, which do bind ATP, but do not hydrolyse it. For this, catalytic residue/s can be mutated in order to make the protein inactive; still, one must take care that the protein retains its native fold. For BmrA, and also for other ABC transporters, it was shown that the mutation of the catalytic glutamate (E504 in BmrA) does not significantly affect the conformational change occurring upon nucleotide binding [23,99]. In contrast, the protein cannot achieve the pre-hydrolytic conformation when the nucleotide-binding Lys residue of the Walker A motif is mutated, here K380A [23,99]. We incubated the mutant E504A with ATP, and then sedimented it for analysis in the solid-state NMR rotor. While E504A is not completely inactive, it displays a very low ATPase activity (but still even crystals were obtained recently, PDB accession code 6R72, and a cryo-EM based structure was reported, PDB accession code 6R81) when compared to K380A, used as a fully inactive control (Figure 5A). After 40 h, only 50% of ATP is consumed, which allowed for the acquisition of 1D and 2D solid-state NMR experiments. The resulting 1D ^31^P CP spectrum displays three narrow peaks corresponding to the three phosphate groups from the ATP bound to the protein (Figure 5B). The 2D ^13^C-^13^C DARR spectrum displays CSPs and peaks appearing, both induced by the conformational and dynamic changes in the protein as a consequence of ATP binding (Figure 5C) [92]. 

To summarize, our data show that analysis of the pre-hydrolytic state is difficult both for DnaB and BmrA, since first the corresponding ATP mimics do not behave in a homogenous manner, i.e., analogues which should yield similar states lead to different NMR spectra, and second, most popular analogues are actually hydrolysed by the helicase in presence of DNA, as well as by the ABC transporter.

With respect to the first point, the intriguing observation that the AMPPCP- and AMPPNP-induced pre-hydrolytic states show conformational differences might be linked to the proposition that one can further differentiate each pre-hydrolytic mimic, as discussed by Ogawa et al., and assign the different mimics to specific steps therein: ATP-γ-S for the initial pre-hydrolysis state, AMPPCP for the pre-isomerization state, ADP:BeF_x_ for the middle pre-hydrolysis state and AMPPNP for the late pre-hydrolysis state [107]. It is difficult to establish a similar statement for DnaB, as one can also explain these differences by the fact that these analogues can behave differently from ATP in terms of their chemical properties: as examples for AMPPNP the oxygen, a hydrogen bond acceptor, is replaced by an NH_2_ group, a possible hydrogen bond donor; AMPPCP has one oxygen atom less than ATP.

With respect to the second point, in the DnaB-DNA complex, only AMPPCP resisted to hydrolysis, and was the best choice to study DnaB and its DNA complex. It was however, rapidly hydrolysed in BmrA, which might be caused by the very high ATPase activity of BmrA, which is with an activity of 6.5 µmol·min^−1^·mg^−1^ one of the most active ABC transporters (one to three orders of magnitude higher than typical ABC transporters) [108]. Amongst AMPPNP and ATP-γ-S, which are the most used pre-hydrolytic state analogues for ABC transporters and helicases (see Figure 3B), neither proved useful here. Alternative strategies using mutant forms were successful to analyse a pre-hydrolytic mimic of the protein and presents a valuable alternative when ATP analogues fail to mimic the pre-hydrolysis states.

### 4.2. The Transition-State Analogues ATP/ADP:Vi and Aluminium Fluorides (ADP:AlF_x_)

In order to investigate the transition states of BmrA and DnaB, we used the solid-state NMR techniques already described above, and also complemented them by EPR (Figure 1). The conformation of DnaB in the presence of ADP:Vi was compared with DnaB apo (Figure 6A) and DnaB in the presence of ADP only (Figure 6B). We also studied the protein with ADP and DNA, in the presence or absence of vanadate (Figure 6C).

The ^13^C-^13^C 2D DARR spectra of DnaB apo and DnaB:ADP:Vi display a few shifting resonances upon binding of the nucleotide (Figure 6A). However, the comparison of DnaB:ADP with and without vanadate shows that the NMR fingerprints of both samples are actually highly similar (Figure 6B), indicating that vanadate did not bind to the NBD and did not induce significant conformational changes. In contrast, when DNA is added to both samples, the NMR spectra of DnaB:ADP:Vi+DNA are different from the ones in the absence of DNA (DnaB:ADP:Vi and DnaB:ADP), with significant CSPs, but the most obvious CSPs are observed for the complex DnaB:ADP:DNA (Figure 6C, left panel). Since these two samples behave differently, a ^31^P NMR spectrum was recorded to probe the bound ATP-mimics. The 1D ^31^P-CP spectrum of DnaB:ADP:DNA displays two phosphorus peaks assigned to DNA (two DNA nucleotides bind to one DnaB monomer leading to two different phosphate binding environments [82]), and four peaks which can be assigned to bound ADP [88] (Figure 6C, right panel). Pα and Pβ correspond to the DnaB:ADP complex in the absence of DNA, and Pα’ and Pβ’ to the DnaB:ADP:DNA complex, indicating an insufficient DNA concentration to saturate the protein completely with DNA. However, the 1D ^31^P-CP spectrum of DnaB:ADP:Vi:DNA (Figure 6C, right panel) shows only one population of ADP, with ^31^P chemical-shift values similar to the DnaB:ADP complex, and a reduced intensity of the peaks assigned to the DNA. One can conclude from these spectra that the presence of vanadate actually inhibits the binding of DNA to DnaB.

For the ABC transporter BmrA, one must say beforehand that conformational changes are not observed in BmrA upon incubating with vanadate and ADP. The protein requires vanadate and ATP instead of ADP, since ATP hydrolysis is required to induce the conformational changes. The inorganic phosphate is then exchanged by a vanadate anion, and Vi and ADP remain bound. The comparison between the ^13^C-^13^C DARR spectra of BmrA apo and BmrA:ATP:Vi shows both CSPs and new appearing peaks (Figure 6D). The presence of additional signals appearing in the protein spectra is indicative of a decrease of the flexibility of the corresponding protein residues [92]. When compared to the pre-hydrolytic state (obtained with BmrA-E504A:ATP), the new peaks appearing in the spectra overlay to a large extent (Figure 6E), indicating that these residues show a similar conformation. Some differences with respect to the pre-hydrolytic state can be observed, which can be associated to the addition of vanadate. To highlight the effect of vanadate, the spectrum of BmrA:ADP:Vi was compared to BmrA:ADP only (Figure 6F). This revealed the presence of new peaks, but only minor CSPs. The appearing peaks can serve as the fingerprint pattern that allows to distinguish the pre-hydrolytic and transition states, while the CSPs serve as the fingerprint pattern reflecting the kind of nucleotide bound. 

A 1D ^31^P-CP NMR experiment can yield complementary information about the bound ATP-mimics (Figure 6F, right panel). The ^31^P spectrum of BmrA:ADP shows the presence of two populations of ADP (labeled Pα, Pβ and Pα’, Pβ’), and the presence of vanadate induces ^31^P chemical-shift changes for BmrA which were not observed for DnaB [12]. In case of BmrA:ADP:Vi, two populations of Pβ can be clearly distinguished and also for Pα, but less significantly (Pβ of ADP with vanadate has a different chemical shift than Pβ of ADP without vanadate). It is known that the trapping of one nucleotide during the transition state (in presence of vanadate) is possible while the second nucleotide can be poorly bound. This property has been observed for several ABC transporters (p-gp [109]; BmrA [99]; LmrA [110]; Maltose transporter [53]) suggesting an asymmetry of the NBDs [111].

In order to gain additional insight into whether vanadate binding occurred or not, we performed EDNMR experiments. This approach can be used to detect the ^51^V nucleus (I = 7/2) in proteins in which the Mg^2+^ has been replaced by the EPR-active Mn^2+^ metal ion [16,112] in the nucleotide-binding sites, as sketched in Figure 1. The experiment detects the hyperfine couplings of the unpaired electrons of Mn^2+^ to the nuclei in the vicinity. We applied this both to the ABC transporter and the DnaB helicase. One should mention that it was shown by biochemical investigations for both proteins that upon substitution of Mg^2+^ by Mn^2+^, their biological function is maintained [86,113]. Figure 7 shows the resulting EDNMR spectrum for the BmrA:ADP:Vi complex (shown in red) with an intense resonance for ^51^V (for the echo-detected field-swept EPR spectra see Appendix A). In the absence of protein in the sample (black line) the spectrum only shows a ^51^V peak with very low intensity assigned to vanadate in solution. Unresolved couplings to ^23^Na would appear at very similar frequencies. We thus conclude that vanadate binds to the NBD in the case of BmrA.

For DnaB, no ^51^V peak can be observed in the EDNMR spectrum, indicating that no vanadate is found in the vicinity of Mn^2+^ (Figure 7B). The EDNMR spectrum indeed shows the same profile for DnaB in the presence of nucleotide with vanadate (red line) and without (black line). We can thus exclude the presence of vanadate in the NBD of the protein. However, as shown previously in Figure 6B, some spectral differences (mainly appearing peaks upon ADP:Vi incubation) can be noticed when DnaB:ADP:Vi was compared to DnaB:ADP. In other words, these experiments do not allow to exclude that vanadate might bind at another location than in the NBD.

We thus used a complementary experiment which can directly detect ^51^V using solid-state NMR. ^51^V has been intensively studied by solid-state NMR due to its rather small nuclear quadrupole moment and its high sensitivity [114,115,116]. Vanadate has also been studied in biological systems using solution-state NMR [117,118] and solid-state NMR [119]. Figure 7C shows the ^51^V MAS spectrum of DnaB:ADP:Vi recorded at two different MAS spinning frequencies of 17 and 19 kHz. By measuring at two different MAS frequencies, the central transition (|−1/2> ↔ |+1/2>, to first order free from quadrupole interaction, can be distinguished from the spinning-sideband positions resulting from first-order quadrupolar interaction (a superposition of the remaining single-quantum transitions, marked by asterisks). The presence of the first order quadrupolar coupling sideband pattern already points to immobilized ^51^V species. We can distinguish two resonances at around −600 ppm (−604 ppm and −618 ppm) and two further vanadate species bound to DnaB at −533 and −681 ppm (Figure 7C). To assign those resonances, a spectrum of the not immobilized (the supernatant) ^51^V was recorded and assigned (Figure 7D). The resonances of the ^51^V MAS spectrum can be assigned by comparison with the solution-state spectrum of the supernatant as follows: VO_4_^3−^ (V1), V_2_O_7_^4−^ (V2), V_4_O_12_^4−^ (V4) and V_5_O_15_^5−^ (V5) [120,121]. We can exclude that the peaks corresponding to the immobilized phase peaks result from the precipitation of vanadate, since with an initial orthovanadate concentration of 5 mM (0.92 g L^−1^) at pH 6, we are two orders of magnitude below the solubility limit. The detected signal thus must stem from DnaB-bound vanadate, which might be related to the observation that addition of vanadate interferes with DNA binding to DnaB (Figure 6C).

To sum up, vanadate is a reasonable ATP-transition-state mimic for the ABC transporter BmrA. The transporter is trapped, most likely in its outward-facing state, when binding ADP:Vi. The transition state is characterized by a characteristic fingerprint in the NMR ^13^C-^13^C DARR spectrum, and vanadate is indeed present in the vicinity of the metal ion. In contrast, ADP:Vi is not a suitable ATP-transition-state mimic for the helicase DnaB. Indeed, solid-state NMR and EPR experiments reveal that vanadate does not bind to the NBD together with the nucleotide. Instead, vanadate is bound elsewhere to DnaB, most likely in an unspecific manner. ADP:Vi strongly inhibits binding of DNA, suggesting that they share the same binding site on DnaB, and that vanadate outcompetes DNA.

### 4.3. Aluminium Fluorides (AlF_x_) as Transition-State Mimic

AlF_x_ is the most frequently used transition-state analogue (Figure 2A), although the pH-dependence of its formation imposes certain limitations to it. At pH ≥ 5 (depending also on the concentration and the anions in the solution), Al^3+^ starts to form an aluminium hydroxide complex, Al(OH)_3,_ which is insoluble. However, the presence of an excess of fluoride shifts the pH upon which Al(OH)_3_ formation occurs to a higher value. We calculated the concentrations of the different species of aluminium under the conditions used (6 mM of AlCl_3_ and 30 mM NH_4_F) as a function of the pH-value (Figure 8). In our case, the formation of Al(OH)_3_ starts at pH 7, and almost all Al^3+^ precipitates as Al(OH)_3_ at pH ≥ 8. The amount of formed AlF_X_ is thus not sufficient to induce the protein:AlF_x_ complex formation. At the same time, fluorides present in the solution can form a complex with Mg^2+^ generating the transition-state analogue MgF_3_^−^. This effect was followed and confirmed by ^19^F NMR for the conversion of a protein:ADP:AlF_4_^−^ complex to a protein:ADP:MgF_3_^−^ complex by increasing the pH [56]. Moreover, as pointed out above, MgF_3_^−^ and AlF_3_ are structurally very similar and some structures comprising AlF_3_ as transition state mimic are in reality MgF_3_^−^ because they were obtained at pH ≥ 8 [56,57,58].

For DnaB, the DnaB:ADP:AlF_x_ complex can easily be prepared at a pH of 6, since the protein is stable at this pH value. In the presence of the transition-state analogue, the 1D CP ^31^P spectrum displays two very narrow resonances assigned to the Pα and Pβ of ADP in complex with AlF_x_ (Figure 9A, left panel). Note a minor amount of DnaB:ADP in the sample. The 2D ^13^C-^13^C DARR spectrum of DnaB:ADP:AlF_x_ displays strong CSPs attributed to conformational changes of the protein (Figure 9A, right panel). While we noticed that the use of vanadate inhibits the binding of DNA, DNA clearly binds to DnaB in the presence of AlF_x_, as shown by Figure 9B, left panel. Fluorescence anisotropy measurements revealed that the affinity for DNA-binding is even the highest in the presence of ADP:AlF_x_ compared to the other ATP-mimics used [82]. The 2D ^13^C-^13^C DARR spectrum of the sample in presence of DNA reveals that several peaks, which belong to the *N*-terminal domain, are again missing, indicating a change in the dynamics of the protein, as was already observed for DnaB:AMPPCP without DNA.

The case of BmrA is more complex, since the optimal pH for sample preparation lies at 8. For optimal use of AlF_x_, the pH would need to be lowered, but we observed this to result in poor (e.g., strongly broadened) spectra. Nevertheless, we explored this further, and in order to test the pH dependency, BmrA, in the presence of ATP, was incubated with 6 mM of AlCl_3_ and 30 mM NH_4_F at pH 8, 7.5 and 7, and a 1D CP ^31^P NMR was recorded for all three conditions (Figure 9C, left panel). The 1D CP ^31^P NMR spectrum at pH 8 shows that ATP/ADP is abundantly co-precipitated with Al(OH)_3_ which makes the 1D spectrum difficult to analyse due to a broad and rather unstructured resonance of this amorphous species (Figure 9C, left panel). As expected, the fraction of ATP/ADP co-precipitated with Al(OH)_3_ decreases with decreasing pH-values. While at pH 7 the precipitation of Al(OH)_3_ is still visible in the ^31^P NMR spectrum, one can compare it to BmrA:ADP:Vi, which shows that both spectra overlay with only few minor differences (Figure 9D, right panel). This indicates that the conformation is highly similar to the one observed with vanadate. A 2D ^13^C-^13^C DARR spectrum recorded on the pH 7 sample (Figure 9C, right panel) confirms this, as the resonances largely superimpose.

To summarize, the use of AlF_x_ as a transition analogue heavily depends on the optimal pH value of the protein. Indeed, biological systems are principally studied at pH 5–9, and it is important to take into account the formation and precipitation of Al(OH)_3_ at pH ≥ 7 under our conditions. The spectra of BmrA at pH ≥ 7 well illustrate the consequences of the use of AlF_x_ in alkaline conditions. The protein actually adopts a similar conformation as in the presence of vanadate, but high amounts of amorphous species are detected. In contrast, DnaB at pH 6 shows high affinity to AlF_x_ which induces substantial conformational changes; and also DNA binding is not affected.

### 4.4. The Post-Hydrolytic State Induced by ADP

The last state in the ATP hydrolysis cycle is the post-hydrolytic state, where ADP is still bound to the protein and the inorganic phosphate (previously γ-phosphate) is released from the binding pocket. This state is well mimicked by the addition of ADP. We used the ^31^P and ^13^C experiments described above to characterize BmrA and DnaB in the presence of ADP.

The ^31^P CP spectrum for DnaB:ADP is shown in Figure 10A (left panel). The spectrum displays two sharp peaks, which indicates a good homogeneity of the sample. The overlay of the 2D DARR spectra DnaB:ADP and DnaB apo (Figure 10A, right panel) reveals CSPs and also the disappearance of *N*-terminal domain peaks, indicating conformational changes and an increase in the dynamics of the protein.

In contrast, the conformational changes of the ABC transporter BmrA are minor between the presence and absence of ADP. First of all, the ^31^P CP spectrum shows the presence of two populations of ADP as identified by peak doubling, labeled Pα, Pβ, and Pα’, Pβ’ (Figure 10B, left panel). These two populations are the result of two different binding modes of ADP to the protein. However, since the intensity of the Pα’, Pβ’ peaks is 50% lower than Pβ, Pα, there is less Pα’, Pβ’ bound to the protein than Pβ, Pα. This is reminiscent to the pattern that was observed with vanadate (Figure 6F). Unspecific binding of ADP to the protein can explain this observation. Secondly, the overlay of the 2D DARR spectra of BmrA:ADP and BmrA apo displays few CSPs compared to what we observed in the presence of other ATP analogues. In contrast to DnaB, the binding of ADP does not induce large conformational or dynamics changes in the protein, and binding of ADP to BmrA seems to be very weak.

### 4.5. Structural Considerations

It seems worthy to compare the NMR data with structural information as available for proteins that are closely related to DnaB of *Helicobacter pylori* and BmrA of *Bacillus subtilis*, respectively.

*BmrA and its structural counterpart.* The most suited for comparison appears to be the set of structures that shows the maltose ABC-transporter of *E. coli* (MBP-MalFGK2) in the outward-facing conformation, with two interacting NBDs and in the presence of AMPPNP (PDB 3RLF), ADP:BeF_3_ (PDB 3PUX), ADP:Vi (PDB 3PUV), and ADP:AlF_4_^-^ (PDB 3PUW) [64]. The collection of these different structures shapes the view of the transport cycle [122]. Chen and Oldham noticed that, despite the different ATP-analogues used, all residues within the NBD are essentially superimposable. However, structural differences between the pre-hydrolytic state (AMPPNP) and the transition state (ADP:Vi and ADP:AlF_4_^−^) are (i) the distance between the γ-phosphate or the mimicked γ-phosphate by the analogues and the bridging oxygen of the β-phosphate and (ii) the presence of a water molecule, essential for the ATP hydrolysis, only in the transition state. Although the transmembrane part of the maltose transporter essentially differs from that of BmrA, the NBD homodimers of the two proteins are relatively similar (RMSD of 1.7 Å from the alignment of MBP-MalFGK2:AMPPNP, PDB 3RLF, with BmrAE504A:ATP, PDB 6R72). The two NBDs differ mainly in their ATPase activity. The ATPase activity of MBP-MalFGK2 is one order of magnitude lower than BmrA [124]. For BmrA, a major conformational transition between the open (inward-facing) and closed (outward-facing) conformation was for example experimentally demonstrated by hydrogen/deuterium exchange (HDX) coupled to mass spectrometry [125] and NMR spectroscopy [92]. It is believed that the protein adopts the closed conformation, with interacting NBDs, upon substrate binding. Generally, in membrane transporters, the energies of their sub-conformations should be close to each other and should essentially depend on the protein environment. The NMR spectra of BmrA in the presence of ADP:Vi and ADP:AlF_4_^−^ might be taken as reporters of the enzyme transition state; in the presence of AlF_4_^−^, the crystal structure of the maltose transporter shows a classical picture with the catalytic water molecule in the apical attack position (Figure 11A). Even in the presence of AlF_4_^-^, the ^31^P 1D CP spectra give two signals for the α- and β-phosphates (Figure 9D), respectively, which points to a certain nonequivalence of the two substrate-binding sites in the two similar NBDs. This finding might indicate that the two catalytic sites operate not simultaneously but sequentially.

*DnaB of Helicobacter pylori and its structural counterparts.* The conformation of DnaB, as could be judged from the 2D ^13^C-^13^C DARR spectra, essentially depends on the nature of the analogue used, which matches the great structural variability reported for DnaB from other bacteria and their viral homologues [126,127,128,129,130,131,132]. Depending on the presence of substrate analogues and their nature, the SF4 helicase subunits can either form rings of distinct shapes [126,127,128,129,130] or arrange themselves as a hexameric ladder along a DNA strand [131,132]. The latter type of the structure was reported for DnaB from *Bacillus stearothermophilus* (currently *Geobacillus stearothermophilus*)*,* which was crystalized, in the presence of a DNA strand, with GDP:AlF_4_^−^ in five of its six catalytic sites [131] (see Figure 11B). In this structure, each monomer of DnaB interacts in a similar way with two nucleotides of DNA; together, the subunits make a kind of a spiral ladder. It is noteworthy, that the position of AlF_4_^−^ in the structure of *Geobacillus stearothermophilus* DnaB (Figure 11B) differs from that in other P-loop fold NTPases. No catalytic water molecule is present apically to the plane of AlF_4_^−^ (see Figure 11A as a typical example), and the position of the AlF_4_^−^ moiety does not correspond to that of the γ-phosphate group (see Figure 11C,F). Interestingly, the NMR data on DnaB from *Helicobacter pylori* discussed herein point to a full occupation of all six NBDs and a rather high symmetry in the oligomer [82] as it potentially could be achieved by more flat conformations of the helicase hexamer, as reported for several DnaB proteins, including the one from *Geobacillus stearothermophilus*, which were crystallized in the absence of AlF_4_^−^ [127,128]. Whether the physiological shape of the DnaB ring is flat or spiral has to be established yet.

Figure 11D,E show the structures of the NBD of the ABC transporter MBP-MalFGK2 (Figure 11D) and the gene 4 helicase from bacteriophage T (Figure 11E) complexed with the pre-hydrolytic ATP analogue AMPPNP. The overlay of the structures (Figure 11F) shows a similar conformation of the bound phosphate chain of the ATP mimic. Although quite similar enzymes seem to bind ATP mimics in a similar way, they might behave differently to the huge number of ATP analogues available and solid-state NMR seems to be the method-of-choice to address such different behaviors.

## 5. Conclusions

We herein reviewed magnetic-resonance approaches (in combination with additional data) to provide information at the atomic level on the binding of mimics of the different ATP forms present during the hydrolysis cycle. We investigated this for two ATP-fuelled proteins, an ABC transporter and a DNA helicase (Table 2), both driven by an ATPase motor domain. We showed that the ATP analogues mainly used for structural studies for such systems, AMPPNP and ATP-γ-S, are not suitable for the systems studied here, since both are hydrolysed by the proteins. Furthermore, we show that analogues which should induce the same state in the hydrolysis cycle can fail to do so, since they result in different conformations. We also discuss that some analogues can interfere with protein function, such as DNA binding for DnaB. NMR, and also EPR, are sensitive tools to assess the impact of different analogues for a given protein, a need that arises through the observation that they can have widely differing effects on different proteins. NMR spectroscopy could be of help in tracing minor differences both in the overall protein conformation and in the state of the phosphate groups. Here we showed that solid-state NMR enabled revealing notable differences in the structural properties of closely related P-loop fold NTPases, namely the SF4 DnaB helicase and BmrA ABC-transporter, which both belong to the same division of ASCE-NTPases.

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
