# Peer review of "ATP Analogues for Structural Investigations: Case Studies of a DnaB Helicase and an ABC Transporter"

_molecules, 2020, doi:10.3390/molecules25225268_

Round 1
Reviewer 1 Report
This paper reviews ATP transition state mimics and their effect on ATP-binding protein conformations using NMR and EPR. The authors compiled data from existing studies as well as original experiments for eight different ATP analogues and analyzed their effect on two proteins from different organisms, DnaB and BmrA. Overall, this is a well-written and in-depth review of existing ATP analogues and their suitability for magnetic resonance-based analyses. Hence, I recommend a minor revision. Below are some comments to consider:
- In figure 4F, the assignment of Pβ and Pγ is rather unconvincing, since the signal-to-noise ratio is quite low. Is it possible to obtain a lower-noise spectrum?
- It was noted that AMPPNP and AMPPCP induced different conformational changes, despite both being pre-hydrolytic state mimics. Is this difference significant enough to the point that these two analogues cannot be interchangeable in experiments? Does this observation overturn the conclusion of any previous studies that used AMPPNP and AMPPCP?
- For the analogues that were found to hydrolyze quickly during analysis, is this phenomenon largely protein dependent, or does this indicate that these analogues are simply not suitable for NMR analysis? Have the authors tried any techniques for delaying hydrolysis?
Author Response
This paper reviews ATP transition state mimics and their effect on ATP-binding protein conformations using NMR and EPR. The authors compiled data from existing studies as well as original experiments for eight different ATP analogues and analyzed their effect on two proteins from different organisms, DnaB and BmrA. Overall, this is a well-written and in-depth review of existing ATP analogues and their suitability for magnetic resonance-based analyses. Hence, I recommend a minor revision. Below are some comments to consider:
- In figure 4F, the assignment of Pβ and Pγ is rather unconvincing, since the signal-to-noise ratio is quite low. Is it possible to obtain a lower-noise spectrum?
We agree with the reviewer that the signal-to-noise ratio (SNR) for the membrane protein BmrA (embedded in lipids) is not as good as for DnaB, despite the long acquisition time of each of the 31P spectra in Figure 4 (21 hours). Improving significantly on SNR would therefore take a significant amount of spectrometer time and cannot be done within the revision timeframe.
2. It was noted that AMPPNP and AMPPCP induced different conformational changes, despite both being pre-hydrolytic state mimics. Is this difference significant enough to the point that these two analogues cannot be interchangeable in experiments? Does this observation overturn the conclusion of any previous studies that used AMPPNP and AMPPCP?
We think no. For the proteins investigated here, the pre-hydrolytic analogues AMPPNP and AMPPCP behave differently and thus cannot be interchanged. The most striking difference for DnaB is that AMPPNP is hydrolysed in presence of DNA, while this was not observed for DnaB alone. This seems to us a significant difference. For other proteins this might be different.
3. For the analogues that were found to hydrolyze quickly during analysis, is this phenomenon largely protein dependent, or does this indicate that these analogues are simply not suitable for NMR analysis? Have the authors tried any techniques for delaying hydrolysis?
The main limitation of NMR is that it takes a couple of hours to prepare (sedimentation) and to measure (see also point 1) the sample and of course, hydrolysis can occur during these processes. The kinetics may be very protein dependent and the behaviour should thus be investigated for each protein and condition used. Many crystal structures were obtained with AMPPNP and hydrolysis seems not to be an issue in these cases. In case of BmrA, ATP hydrolysis can be suppressed by introducing point mutations in the protein, e.g. by mutating the catalytic glutamate. We did not try other approaches.
Reviewer 2 Report
- It would be appropriate to explain in the abstract why DnaB helicase and the ABC transporter BmrA were selected.
- It would be appropriate to include the explanation of ATP and GTP abbreviations in the Introduction.
- Nuclear magnetic resonance is written in capital letters in the introduction and in lower case in the abstract.
- The abbreviation EPR is not explained in the introduction.
- Authors could include a table summarizing the employment of the different magnetic-resonance approaches cited in Figure 1. Although they are mentioned in the text, the incorporation of a table would help a better understanding.
- At the end of the first paragraph it is mentioned in parentheses “as shown in this work for the two model systems discussed”. The author should include this statement in the text without parentheses. Please, avoid using parentheses for clarifications.
- Authors must check the legends of the figures and unify formats.
Overall comment: The manuscript is well written and the work is complete however, sometimes, the reading is complicated. Authors could improve this issue removing details that are not relevant and/or using more tables.
Author Response
- It would be appropriate to explain in the abstract why DnaB helicase and the ABC transporter BmrA were selected.
We actually started the two projects in an independent manner, but soon realized that there were many common points to them, as they indeed belong to the same group of P-loop fold NTPases. We have mentioned that now briefly in the abstract.
“The proteins investigated are the bacterial DnaB helicase from Helicobacter pylori and the multidrug ABC transporter BmrA from Bacillus subtilis, both belonging to the same division of P-loop fold NTPases.”
2.It would be appropriate to include the explanation of ATP and GTP abbreviations in the Introduction.
We have included it.
3. Nuclear magnetic resonance is written in capital letters in the introduction and in lower case in the abstract.
We have corrected that.
4.The abbreviation EPR is not explained in the introduction.
We have added the abbreviation in the introduction.
5. Authors could include a table summarizing the employment of the different magnetic-resonance approaches cited in Figure 1. Although they are mentioned in the text, the incorporation of a table would help a better understanding.
We thank the reviewer for his suggestion and have included a new Table 1 in our manuscript summarizing the magnetic-resonance techniques applied and their information content.
6. At the end of the first paragraph it is mentioned in parentheses “as shown in this work for the two model systems discussed”. The author should include this statement in the text without parentheses. Please, avoid using parentheses for clarifications.
We have modified that.
7.Authors must check the legends of the figures and unify formats.
We have carefully checked the format of the figure legends and unified it.
Overall comment: The manuscript is well written and the work is complete however, sometimes, the reading is complicated. Authors could improve this issue removing details that are not relevant and/or using more tables.
Reviewer 3 Report
This manuscript presented the lessons learned on two distinct ATPase DnaB and BmrA from the binding of mimics of the different ATP forms presented during the hydrolysis cycle. The authors reviewed the magnetic-resonance strategies that can be of use to prove the binding of the ATP-mimics, and presented C-13, P-31 and V-51 solid-state NMR spectra of the proteins or the bound molecules to unravel conformational and dynamic changes upon binding of the ATP-mimics. As a review paper, the authors also discussed which analogs best mimic the different hydrolysis states for the DnaB helicase and the BmrA as well as some analogs that can interfere with the protein function such as DNA binding for these two ATPase. Additionally the authors presented some original data on solid state NMR experiments and EPR experiments, which helps their conclusions. Generally this is well-organized review. I do have a few of questions for the authors though.
- While the authors used ATP analogs to mimic the pre-hydrolytic state, transition-state and poste-hydrolytic state, we can see the changes in the NMR spectra. But we don't have any information about how this NMR spectra information is related to the binding and the hydrolytic mechanism. There are some explanations of the NMR spectra changes but not enough.
- For AMPPNP and AMPPCP in the pre-hydrolytic state, both as ATP mimics binding to ATPase in the ATP binding pocket, their differences in the NMR spectra and the results from these differences are not well explained.
- In the pre-hydrolytic state, the authors claimed that AMPPCP and AMPPNP can be hydrolyzed by BmrA. It might be the case for AMPPNP, but it is difficult to believe that AMPPCP can also be hydrolyzed because from the mechanism, you will need to break a P-C bond for this hydrolysis reaction to finalize and it does not look possible.
- Also the authors indicated that the binding of ADP to BmrA seems to be very weak, which should be the case according to the reaction mechanism. But the ADP should also have weak binding for DnaB and this is not indicated in the paper. If ADP has reasonably binding affinity towards DnaB, what is the cause?
- On page 11, 2nd paragraph, 7th line from the end of the paragraph, "(Figure 4C, right panel)" should be "(Figure 6C, right panel)"?
Author Response
This manuscript presented the lessons learned on two distinct ATPase DnaB and BmrA from the binding of mimics of the different ATP forms presented during the hydrolysis cycle. The authors reviewed the magnetic-resonance strategies that can be of use to prove the binding of the ATP-mimics, and presented C-13, P-31 and V-51 solid-state NMR spectra of the proteins or the bound molecules to unravel conformational and dynamic changes upon binding of the ATP-mimics. As a review paper, the authors also discussed which analogs best mimic the different hydrolysis states for the DnaB helicase and the BmrA as well as some analogs that can interfere with the protein function such as DNA binding for these two ATPase. Additionally the authors presented some original data on solid state NMR experiments and EPR experiments, which helps their conclusions. Generally this is well-organized review. I do have a few of questions for the authors though.
- While the authors used ATP analogs to mimic the pre-hydrolytic state, transition-state and poste-hydrolytic state, we can see the changes in the NMR spectra. But we don't have any information about how this NMR spectra information is related to the binding and the hydrolytic mechanism. There are some explanations of the NMR spectra changes but not enough.
Changes in NMR spectra indicate in general a modification of the electronic environment of the nuclei. For 13C nuclei shift changes are often dominated by a modification in dihedral angles, and thus conformation. How the differences relate exactly to structure is however difficult to establish; NMR thus reveals with high sensitivity that conformational changes occur, but its weak point is that relating them to the structural changes by which they are caused, and thus the underlying mechanism, is far from straightforward.
2. For AMPPNP and AMPPCP in the pre-hydrolytic state, both as ATP mimics binding to ATPase in the ATP binding pocket, their differences in the NMR spectra and the results from these differences are not well explained.
See answer to point (1) – we know that they are different, since the chemical shifts are different, but do not know in which respect they are different.
3. In the pre-hydrolytic state, the authors claimed that AMPPCP and AMPPNP can be hydrolyzed by BmrA. It might be the case for AMPPNP, but it is difficult to believe that AMPPCP can also be hydrolyzed because from the mechanism, you will need to break a P-C bond for this hydrolysis reaction to finalize and it does not look possible.
We agree with the reviewer that hydrolysis of AMPPCP between the beta and gamma phosphate is difficult to understand mechanistically. However, the 31P direct pulsed spectra (Figure 4E) clearly indicate hydrolysis over time, possibly the formation of AMP and inorganic pyrophosphate. We have indicated in Table 1 now that hydrolysis is observed, but we are not sure if it is the protein which actually induces it. Therefore, Table 1 now only states “hydrolysis observed”.
4. Also the authors indicated that the binding of ADP to BmrA seems to be very weak, which should be the case according to the reaction mechanism. But the ADP should also have weak binding for DnaB and this is not indicated in the paper. If ADP has reasonably binding affinity towards DnaB, what is the cause?
We have not determined the binding affinities of the different ATP mimics and thus do not have an answer to the question if the KD value for ADP are larger than for the other mimics investigated. Since we are using a rather huge excess, we still were able to saturate all binding sites with ADP, as e.g. also observed in the crystal structure of the AaDnaB helicase with ADP (Strycharska, M. S.; Arias-Palomo, E.; Lyubimov, A. Y.; Erzberger, J. P.; O’Shea, V.; Bustamante, C. J.; Berger, J. M., Nucleotide and partner-protein control of bacterial replicative helicase structure and function. Mol. Cell 2013, 52 (6), 844-854).
5. On page 11, 2nd paragraph, 7th line from the end of the paragraph, "(Figure 4C, right panel)" should be "(Figure 6C, right panel)"?
We thank the reviewer for pointing out that mistake.